# A randomized, double-blind, placebo-controlled pilot trial of low-intensity pulsed ultrasound therapy for refractory angina pectoris

**Tomohiko Shindo**[1], Kenta Ito[1], Tsuyoshi Ogata[1], Ryo Kurosawa[1], Kumiko Eguchi[1], Yuta Kagaya[1], Kenichiro Hanawa[1], Yuhi Hasebe[1], Kensuke Nishimiya[1], Takashi Shiroto[1], Jun Takahashi[1], Yasuo Okumura[2], Teruo Noguchi[3], Yukio Ozaki[4], Hiroyuki Daida[5], Nobuhisa Hagiwara[6], Tohru Masuyama[7], Taishiro Chikamori[8], Yoshihiro Fukumoto[9], Kenichi Tsujita[10], Hiroshi Kanai[11], Satoshi Yasuda[1], Hiroaki Shimokawa[12]¤*

**1** Department of Cardiovascular Medicine, Tohoku University Graduate School of Medicine, Sendai, Japan, **2** Department of Medicine, Nihon University Graduate School of Medicine, Tokyo, Japan, **3** Department of Cardiovascular Medicine, National Cerebral and Cardiovascular Center, Suita, Japan, **4** Department of Cardiology, Fujita Health University School of Medicine, Toyoake, Japan, **5** Department of Cardiology, Juntendo University Graduate School of Medicine, Tokyo, Japan, **6** Department of Cardiology, Tokyo Women's Medical University Graduate School of Medicine, Tokyo, Japan, **7** Cardiovascular Division, Department of Internal Medicine, Hyogo Medical University Graduate School of Medicine, Nishinomiya, Japan, **8** Department of Cardiology, Tokyo Medical University, Tokyo, Japan, **9** Kurume University School of Medicine, Kurume, Japan, **10** Division of Cardiovascular Medicine, Department of Internal Medicine, Kumamoto University Graduate School of Medical Sciences, Kumamoto, Japan, **11** Division of Biomedical Measurements and Diagnostics, Graduate School of Biomedical Engineering, Tohoku University, Sendai, Japan, **12** Department of Cardiovascular Medicine, International University of Health and Welfare, Narita, Japan

¤ Current address: Graduate School, International University of Health and Welfare, Narita, Japan
* shimo@cardio.med.tohoku.ac.jp, shimo@iuhw.ac.jp

**Data Availability Statement:** The data underlying the results presented in the study are available from Department of Cardiovascular Medicine,

## Abstract

### Background

Despite the advances in the treatment of cardiovascular diseases, effective treatment remains to be established to improve the quality of life and prognosis of patients with chronic coronary syndromes. This study was aimed to evaluate the effectiveness and safety of the low-intensity pulsed ultrasound (LIPUS) therapy, which we have developed as a novel non-invasive angiogenic therapy through upregulation of endothelial nitric oxide synthase (eNOS).

### Methods and findings

We conducted a randomized, double-blind, placebo-controlled (RCT) pilot trial of the LIPUS therapy for patients with refractory angina pectoris. The patients who received optimal medical therapy without indication of PCI or CABG due to the lack of graftability or complexity of coronary lesions were enrolled. They were randomly divided into the LIPUS treatment group (N = 31) and the placebo group (N = 25) in a 1:1 fashion. The LIPUS therapy was performed in a transthoracic manner for 20 min for 3 sections each (mitral, papillary muscle, and apex

Tohoku University Graduate School of Medicine (8122-717-7153).

**Funding:** This study was funded by the Japan Agency for Medical Research and Development (AMED) (No. 18lm0203047h0001). The funders had no role in study design, data collection and analysis, decision to publish, or preparation of the manuscript.

**Competing interests:** The authors have declared that no competing interests exist.

levels) under the conditions that we identified; frequency 1.875 MHz, intensity 0.25 MPa, and 32 cycles. The primary endpoint was weekly use of nitroglycerin. Secondary endpoints included stress myocardial perfusion imaging and others. The average weekly nitroglycerin use (times/week) was decreased from 5.50 to 2.44 in the LIPUS group and from 5.94 to 2.83 in the placebo group. The changes in the average weekly nitroglycerin use were comparable; -3.06 (95% CI: -4.481 to -1.648) in the LIPUS group (P<0.01) and -3.10 (95% CI: -4.848 to -1.356) in the placebo group (P<0.01). No adverse effects were noted.

## Conclusions

In the present study, the LIPUS therapy did not further ameliorate chest pain as compared with optimal medications alone in patients with refractory angina pectoris. The present findings need to be confirmed in another trial with a large number of patients. (Registration ID: UMIN000012369).

## Introduction

Percutaneous coronary intervention (PCI) and coronary artery bypass grafting (CABG) are established treatments for acute coronary syndrome [1]. On the other hand, regarding chronic coronary syndrome, current evidence is not still enough for physicians to select a proper treatment strategy [2]. The COURAGE and the ISCHEMIA trials have demonstrated that invasive treatment strategy may provide no additional prognostic benefits on the top of optimal medical therapy for chronic coronary syndrome [3, 4]. These results may be caused by procedure-related complications of PCI or CABG, high-risk of bleeding due to dual-antiplatelet therapy (DAPT) or the presence of coronary microvascular dysfunction that are not ameliorated by PCI or CABG. Especially, patients with refractory angina without indication of PCI or CABG despite optimal medical therapy could be regarded as a high-risk population with poor prognosis and therefore require a further therapeutic option. Furthermore, the ORBITA trial demonstrated that PCI for a single-vessel disease (>70% stenosis in the left anterior descending coronary artery) has no additional benefits for exercise tolerance as compared with placebo group [5]. This study suggests that much of symptomatic improvements after PCI may be due to placebo effects [5]. Thus, effective therapeutic strategy to improve quality of life (QOL) and prognosis of patients with refractory angina remains to be established.

We have developed a low-intensity pulsed ultrasound (LIPUS) therapy as a novel, non-invasive angiogenic therapy, which upregulates endothelial nitric oxide synthase (eNOS) and other angiogenic factors selectively in ischemic tissues [6]. Pulsed ultrasound with specific conditions has been reported to exert therapeutic effects, such as cell proliferation and anti-inflammatory responses [7, 8]. Mechanistically, by physically inducing cyclic strain, LIPUS stimulates caveolae that locate on the surface of endothelial cell membrane [9, 10]. We demonstrated that LIPUS phosphorylates Ser1177 of eNOS under the presence of caveolae, and subsequent phosphorylation of downstream Akt and Erk1/2 upregulates eNOS and VEGF expressions [11]. This therapeutic mechanism is called as "mechano-transduction" [12–14]. Clinical application of the mechano-transduction mechanism could be expected as a safe and highly effective local treatment strategy [14].

In the present study, we conducted a randomized, double-blind, placebo-controlled pilot trial in order to address the efficacy and safety of the LIPUS therapy for patients with refractory angina without indication of PCI or CABG.

## Methods

### Study protocols

We enrolled patients with refractory angina without indication of PCI or CABG despite optimal medical therapy, who had angina episode at least once a week. Specifically, on the premise that existing optimal medical therapy has been performed, we enrolled patients who were found to be difficult to apply PCI or CABG due to the lack of graftability or complexity of lesions at the heart team conference. Angiography was performed in all enrolled patients, and only patients who have evidence of myocardial ischemia by invasive functional evaluation or stress myocardial perfusion imaging were enrolled. They were enrolled in 2013–2018, and were able to regularly visit an outpatient clinic. The registration of this study was extended from 3 years to 6 years because of the interruption of funding source midway through. All the patients were given informed consent and they were obtained by written form. They were randomly and automatically assigned to either the LIPUS group or the placebo group in a 1:1 fashion, according to the following 4 allocation factors; Canadian Cardiovascular Society (CCS) class score (II, III or IV), weekly use of nitroglycerin (4 times a week or less, or 5 times a week or more), left ventricular ejection fraction (40% or more, or less than 40%), and participating hospital. In this trial including 10 participating institutes, because 4 stratification factors were used, our first sample size setting was 80 patients. However, it was redesigned to enroll 60 patients in the middle of the trial due to delays in patient enrollment. In order to avoid any possible pathological angiogenesis, patients with malignancies or uncontrolled diabetic retinopathy were excluded. The diagnosis and approval of enrollment were made by a third-party committee. After we confirmed patient's eligibility, subject information was sent to the registration center, randomized allocation was performed, and then the allocation results were documented to the open-label treatment physicians at each institution. In this study, all participating physicians and supporting staff except un-blinded physician who handled the LIPUS or placebo procedure were blinded. Authors could not access to the information that could identify individual participants during data collection. This study was conducted at 10 major cardiovascular institutes in Japan, including Kumamoto University Hospital, Kurume University Hospital, Hyogo Medical University Hospital, Fujita Health University Hospital, Tokyo Women's Medical University Hospital, Nihon University Hospital, Tokyo Medical University Hospital, Juntendo University Hospital, National Cerebral and Cardiovascular Center, and Tohoku University Hospital. This study was approved by the institutional review board of each institute under the Good Clinical Practice and was conducted with the ethical principles under the Declaration of Helsinki. The protocol of the present study was approved by the institutional review board of the Tohoku University Hospital (No. 133004) and was registered as UMIN Trial ID: UMIN000012369. All the clinical data obtained in this trial can be disclosed upon appropriate request.

### LIPUS therapy

Patients were placed in the left lateral position, and the probe was placed on the anterior chest wall to observe the heart and depict the ischemic site (**Fig 1A**). By changing the setting to the ultrasound therapeutic mode, the depiction range became the treatment range. The ischemic area was determined in advance by invasive functional evaluation or stress myocardial perfusion imaging. Based on the preliminary study, the conditions of the LIPUS therapy in the trial were as follows; scanning beam with sector-type probe, frequency 1.875 MHz, intensity 0.25 MPa, transmission wave number 32 each, pulse repetitive frequency (PRF) 1 kHz, and target depth 6–8 cm [6, 11]. The focus of LIPUS therapy was set at 4–8 cm, which is the depth of a

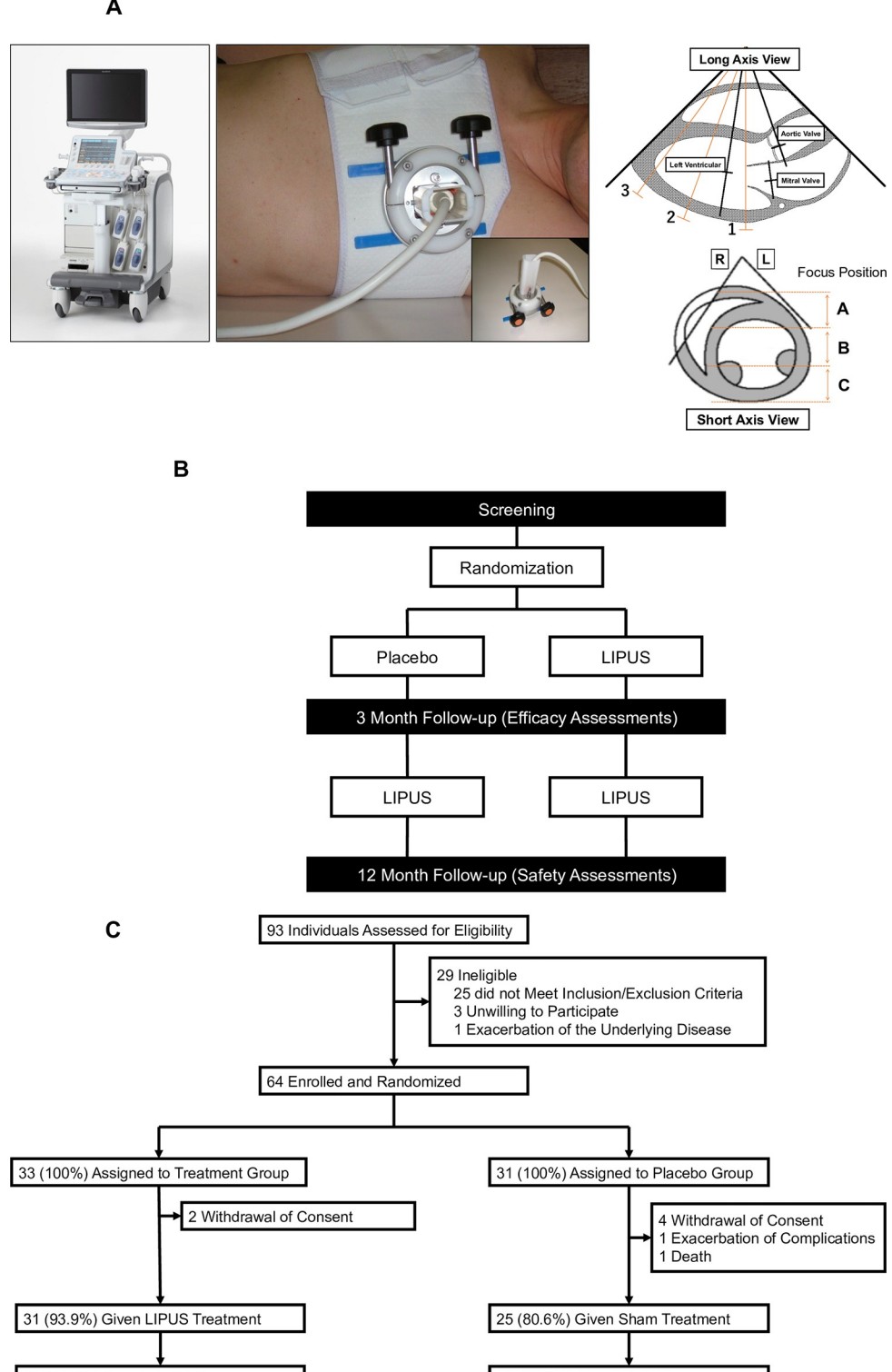

**Fig 1. LIPUS therapy for angina pectoris. (A) Left**, LIPUS therapy equipment (LIPUS machine and chest band). **Right**, 3 cross-sections for the therapy (mitral valve, papillary muscle, and apical levels). **(B)** Trial protocols. **(C)** CONSORT flow diagram. LIPUS: low-intensity pulsed ultrasound.

typical human heart, and the depth of focus was adjusted by the treating physician using diagnostic echocardiography. LIPUS was applied for 20 min per cross-section, and was irradiated for 3 cross-sections from the base to the apex of the heart (mitral valve, papillary muscle, and apex levels) at an interval of 5 min (**Fig 1A**). One session consisted of the LIPUS therapy every other day for 3 days. The study protocol with the first and second series is shown in **Fig 1B**. In the first series, after the screening period, the LIPUS therapy and sham treatment were performed in a double-blind manner for patients and physicians, and only the un-blinded sonographer (physician) knew the therapy option. In the second series, 3 months after the first series, additional LIPUS therapy was applied to all patients. Efficacy outcomes were evaluated at 3 months after the first series, while safety outcome at one year after the second series (**Fig 1B**).

## Endpoints

The primary endpoint was weekly use of nitroglycerin at 3 months after the therapy. The secondary endpoints included chest pain frequency, CCS class score, 6-min walking distance, stress myocardial perfusion imaging, echocardiography, cardiac MRI, and adverse events. Weekly use of nitroglycerin and chest pain frequency were recorded by individual patient in dedicated notebooks received after the registration, and at the follow-up period, collected by blinded clinical research coordinator (CRC). During the study period, the notebooks were reviewed monthly by the CRC to confirm their accuracy regarding chest symptoms and frequency of nitroglycerin use. CCS class score was recorded by another blinded cardiologist. Changes in walking distance of the 6-min walking distance from baseline to 84 days after the initial implementation of treatment were obtained. Changes of left ventricular ejection fraction and abnormal wall motion in the echocardiography and cardiac MRI from baseline to 84 days after the initial implementation of treatment were obtained. As an evaluation of myocardial ischemia by stress myocardial perfusion imaging, the summed difference score (SDS) of the treatment area and the change in the ischemic amount before and 3 months after the treatment were calculated. Ischemic area was determined by prior invasive functional evaluation or stress myocardial perfusion imaging, and the analysis area of the heart was divided into 17 segments, and only the treatment-targeted area was extracted and analyzed [15]. In Table 2, we reported all the changes (delta) of variables between baseline and 84 days after the initial implementation of treatment, including SDS, wall motion abnormalities at rest. Adverse events included all-cause death and major adverse cardiovascular events (MACE) including a composite of cardiovascular death, non-fatal myocardial infarction, and hospitalization due to unstable angina or heart failure. Regarding the analysis of long-term prognosis, we analyzed not only the cases that completed follow-up, but also all the cases that received the therapy at least once in this study.

## Post-hoc analysis

In addition, we evaluated the correlation in stress myocardial perfusion imaging between the extent of baseline myocardial ischemia and post-treatment changes in the ischemia. This correlation analysis was performed to evaluate the effect of the LIPUS therapy when applied twice, between baseline and 6 months after the therapy.

## Statistical analysis

For the statistical analysis of primary and secondary endpoints, the mean value of change before and after the LIPUS therapy and the confidence interval were estimated, and comparisons were made between the 2 groups using the t-test. The Wald method was used for the

comparison of the proportion of improved cases after the treatment and the estimation of the confidence interval. MACE and all-cause death were compared as cumulative incidence using Gray's test where dropout due to death from non-cardiovascular causes or non-MACE adverse events were treated as a competing risk. For the estimation of the hazard ratio and P-value, the Fine and Gray model, which is an extension hazard model of the Cox's proportional hazards model, was used. Point estimates and 95% confidence intervals for adverse event withdrawal rates were calculated for each group at 450 days after the initial treatment. P value less than 0.05 was considered to be statistically significant.

## Results

### Baseline patient characteristics

Baseline patient characteristics are shown in **Table 1**. The average age of the LIPUS group was 70 year-old and that of the placebo group was 73.5 (**Table 1**). Among the 64 patients enrolled in the trial, 33 were assigned to the LIPUS group and 31 to the placebo group (**Fig 1C**). After the randomization, 2 in the LIPUS group did not complete the planned protocol due to withdrawal of consent, while 6 in the placebo group did not complete the trial (exacerbation of complications in 1, withdrawal of consent in 4, and death in 1 due to acute aortic dissection). Thus, 31 patients in the LIPUS group and 25 in the placebo group were analyzed for efficacy and safety endpoints (**Fig 1C**). There were no significant differences in baseline patient characteristics between the 2 groups, expect for the incidence of atrial arrhythmias and diastolic blood pressure.

### Primary endpoint

The average weekly use of nitroglycerin was 5.5 in the LIPUS group and 5.9 in the placebo group, and the average CCS class score was 2.4 in the LIPUS group and 2.6 in the placebo group. The average weekly use of nitroglycerin (times/week) decreased from 5.50 to 2.44 in the LIPUS group and from 5.94 to 2.83 in the placebo group (**Fig 2A**). The changes in the weekly use of nitroglycerin were -3.06 (95% CI: -4.481 to -1.648) in the LIPUS group (P<0.01) and -3.10 (95% CI: -4.848 to -1.356) in the placebo group (P<0.01), with no difference between the 2 groups (P = 0.97). Similarly, there was no significant difference between the 2 groups in the proportion of patients whose average weekly use of nitroglycerin improved (P = 0.14).

### Secondary endpoints

CCS class score and average weekly frequency of chest pain decreased significantly but equally in the 2 groups (**Table 2**). Treadmill exercise test, 6-min walk distance, echocardiography, and cardiac MRI also showed no significant difference between the 2 groups (**Table 2**). Echocardiographic wall motion abnormalities were -0.41 (-2.622–1.802, N = 30) in the LIPUS group and 0.18 (-2.406–2.776, N = 26) in the placebo group, which did not change from pre-treatment level (at the time of screening) in both groups (P = 0.72). Importantly, in stress myocardial perfusion imaging, the average extent of myocardial ischemia decreased from 13.71 to 12.84 in the LIPUS group, but increased from 6.41 to 8.48 in the placebo group (P = 0.18) (**Fig 2B**).

### Post-hoc analysis

This analysis was performed because of the significant baseline difference in the extent of myocardial ischemia seen in the stress myocardial perfusion imaging. However, adjusting the baseline myocardial ischemia in an ANCOVA analysis was not effective because of this small

**Table 1. Baseline patient characteristics.**

| | LIPUS Group (N = 31) | Placebo Group (N = 27) | P-value |
|---|---|---|---|
| Age±SD, years | 70.0±11.3 | 73.5±11.7 | 0.25 |
| Sex | | | |
| Female, n (%) | 6 (19.4%) | 10 (37.0%) | 0.15 |
| Male, n (%) | 25 (80.6%) | 17 (63.0%) | |
| Comorbidities | | | |
| Hypertension, n (%) | 29 (93.5%) | 24 (88.9%) | 0.66 |
| Hyperlipidemia, n (%) | 27 (87.1%) | 23 (85.2%) | 1.00 |
| Diabetes mellitus, n (%) | 22 (71.0%) | 18 (66.7%) | 0.78 |
| Heart failure, n (%) | 9 (29.0%) | 12 (44.4%) | 0.28 |
| Stroke, n (%) | 1 (3.2%) | 3 (11.1%) | 0.33 |
| Myocardial infarction, n (%) | 1 (3.2%) | 0 (0.0%) | 1.00 |
| Valvular disease, n (%) | 3 (9.7%) | 4 (14.8%) | 0.69 |
| PAD/ Aortic aneurysm, n (%) | 10 (32.3%) | 13 (48.1%) | 0.28 |
| COPD/ Respiratory disease, n (%) | 9 (29.0%) | 4 (14.8%) | 0.22 |
| Atrial fibrillation/Arrhythmia, n (%) | 5 (16.1%) | 11 (40.7%) | 0.04 |
| Kidney disease/Renal dysfunction, n (%) | 12 (38.7%) | 11 (40.7%) | 1.00 |
| Digestive disease, n (%) | 7 (22.6%) | 6 (22.2%) | 1.00 |
| Medications | | | |
| Aspirin/P2Y12 inhibitors, n (%) | 31 (100.0%) | 25 (92.6%) | 0.21 |
| Beta-blockers, n (%) | 28 (90.3%) | 24 (88.9%) | 1.00 |
| Calcium channel blockers, n (%) | 20 (64.5%) | 14 (51.9%) | 0.42 |
| RASi, n (%) | 23 (74.2%) | 16 (59.3%) | 0.27 |
| Lipid-lowering therapy, n (%) | 26 (83.9%) | 24 (88.9%) | 0.71 |
| Antidiabetic drugs, n (%) | 15 (48.4%) | 12 (44.4%) | 0.80 |
| SGLT2i, n (%) | 4 (12.9%) | 0 (0.0%) | 0.12 |
| Nitroglycerin/Vasodilators, n (%) | 31 (100.0%) | 27 (100.0%) | - |
| Loop diuretic/$V_2$ receptor antagonists, n (%) | 11 (35.5%) | 10 (37.0%) | 1.00 |
| MRA, n (%) | 2 (6.5%) | 5 (18.5%) | 0.23 |
| Oral anticoagulants, n (%) | 4 (12.9%) | 8 (29.6%) | 0.19 |
| Antiarrhythmic drugs, n (%) | 1 (3.2%) | 3 (11.1%) | 0.33 |
| Digoxin, n (%) | 1 (3.2%) | 0 (0.0%) | 1.00 |
| Vital signs | | | |
| Systolic blood pressure±SD, mmHg | 127.5±18.1 | 123.2±18.4 | 0.38 |
| Diastolic blood pressure±SD, mmHg | 69.7±10.4 | 64.7±7.7 | 0.05 |
| Heart rate±SD, beats per min | 69.3±13.3 | 64.5±11.6 | 0.15 |
| Height±SD, cm | 161.3±7.7 | 158.9±9.4 | 0.29 |
| Body weight±SD, kg | 64.0±14.7 | 62.4±8.6 | 0.62 |
| Body mass index±SD, kg/m$^2$ | 24.4±4.4 | 24.7±3.0 | 0.76 |
| Echocardiography LVEF±SD, % Laboratory data | 53.5±13.2 | 53.1±12.0 | 0.90 |
| Creatinine±SD, mg/dL | 2.4±3.0 | 2.1±2.8 | 0.67 |
| LDL-cholesterol±SD, mg/dL | 80.3±21.8 | 81.5±25.2 | 0.85 |
| HbA1c±SD, % | 6.7±1.0 | 6.7±1.1 | 0.77 |
| CRP±SD, mg/dL | 0.34±0.37 | 0.28±0.39 | 0.61 |
| BNP±SD, pg/dL | 366.2±771.8 | 265.1±422.3 | 0.55 |

Results are presented as means±SD. Categorical variables are presented as counts and percentages. PAD denotes Peripheral Arterial Disease, COPD Chronic Obstructive Pulmonary Disease, RASi Renin-Angiotensin System inhibitors, SGLT2i Sodium Glucose Co-transporter 2 inhibitors, MRA Mineralocorticoid Receptor Antagonists, LVEF Left Ventricular Ejection Fraction, LDL-C Low Density Lipoprotein-cholesterol, CRP C-reactive protein, BNP Brain Natriuretic Peptide.

**Table 2. Clinical outcomes.**

| | LIPUS Group (N = 31) | Placebo Group (N = 27) | P-value |
|---|---|---|---|
| Primary endpoint | | | |
| Weekly nitroglycerin use | -3.06 (-4.481, -1.648) | -3.10 (-4.848, -1.356) | 0.97 |
| Secondary endpoints | | | |
| CCS class score | -0.50 (-0.773, -0.227) | -0.54 (-0.824, -0.253) | 0.84 |
| Chest pain, per week | -3.98 (-5.796, -2.156) | -5.25 (-9.057, -1.443) | 0.52 |
| Treadmill exercise load, kcal | -2.40 (-9.041, 4.233) | 3.87 (-1.982, 9.726) | 0.16 |
| Treadmill exercise load time, min | -0.55 (-1.639, 0.538) | 0.37 (-0.548, 1.295) | 0.21 |
| 6-min walk distance, m | -0.7 (-52.60, 51.20) | 12.3 (-28.12, 52.72) | 0.66 |
| Stress myocardial perfusion imaging (SDS) | 0.31 (-1.149, 1.770) | 0.00 (-1.969, 1.969) | 0.79 |
| Echocardiography LVEF, % | -0.41 (-2.622, 1.802) | 0.18 (-2.406, 2.776) | 0.72 |
| Cardiac MRI LVEF, % | 4.03 (-10.520, 18.573) | 2.98 (-4.109, 10.060) | 0.84 |
| Cardiac MRI stroke volume, mL | -7.1 (-24.63, 10.39) | 13.3 (-1.10, 27.68) | 0.07 |

CCS denotes Canadian Cardiovascular Society, LVEF left ventricular Ejection fraction, MRI Magnetic Resonance Imaging.

sample size. So, we compared the patients with the extent of myocardial ischemia less than 30% of the left ventricle, and we found that myocardial ischemia improved depending on the extent of baseline myocardial ischemia only in the LIPUS group (P<0.01) but not in the placebo group (**S1 Fig**). **S2 Fig** shows representative images of a patient who showed improvement in myocardial ischemia in response to the LIPUS therapy.

## Adverse events

The median follow-up period for both groups was 452 days for the LIPUS group and 449 days for the placebo group. All-cause death was observed in 1 of 32 patients (3.1%) in the LIPUS group and in 4 of 29 patients (13.8%) in the placebo group (95% CI: 0.027 to 1.922, HR 0.23, P = 0.17) (**Table 3**). All-cause death tended to be improved in the LIPUS groups as compared with the placebo group (P = 0.17, HR 0.23) (**Fig 2C**). MACE was documented in 4 of 32 patients (12.5%) in the LIPUS group and in 5 of 29 patients (17.2%) in the placebo group (95% CI: 0.192 to 2.587, HR 0.71, P = 0.60) (**Table 3**). There was no difference in MACE between the 2 groups (P = 0.60, HR 0.71) (**Fig 2D**).

There were 7 device defects occurred during the treatment period (6 in the LIPUS group and 1 in the placebo group), which were immediately repaired. All of the device defects were damage of breast band or fixture which fix the probe with breast band. In this trial, there were no serious health damages related to the LIPUS therapy.

## Discussion

In the present study, we conducted a pilot trial to examine the efficacy and safety of the LIPUS therapy for patients with refractory angina. We found no significant differences in the primary subjective endpoints between the 2 groups. There was no serious adverse effect of the LIPUS therapy. To the best of our knowledge, this is the first clinical trial to address the efficacy and safety of the LIPUS therapy in patients with refractory angina.

In the past clinical trials for chronic coronary syndrome, most of the investigators focused on 2 major outcomes, including frequency of chest pain and long-term prognosis [16]. In terms of frequency of chest pain, although there are some studies showing that PCI and optimal medical therapy (OMT; e.g. beta-blockers) improved QOL of stable angina patients [17], the ORBITA trial clearly demonstrated that the placebo effects are substantially involved in the

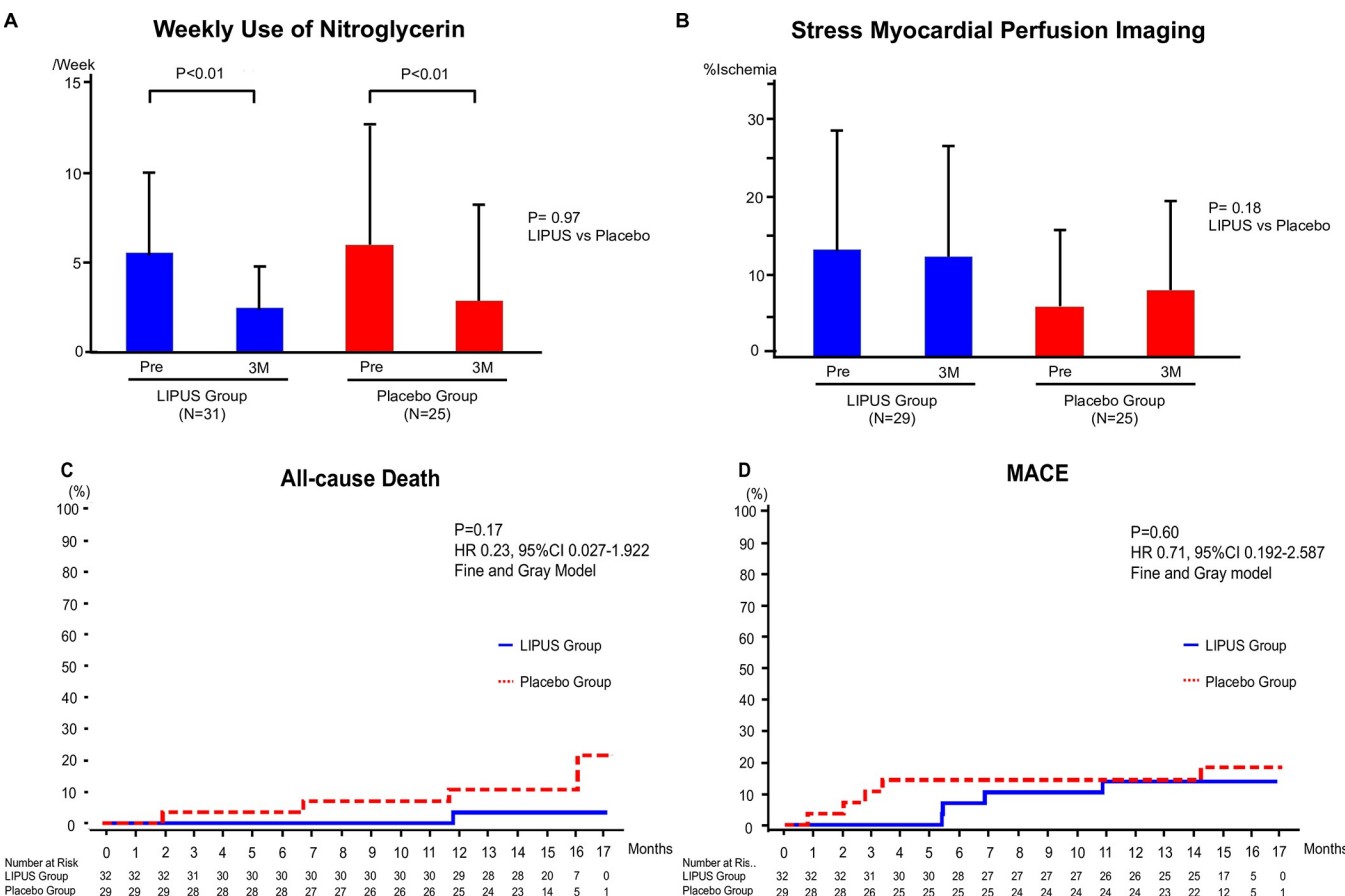

**Fig 2. Results of the LIPUS therapy on primary and secondary outcomes. (A)** Weekly use of nitroglycerin. **(B)** Extent of myocardial ischemia evaluated by stress myocardial perfusion imaging. **(C)** Kaplan-Meier curve of all-cause death. **(D)** Kaplan-Meier curve of MACE. LIPUS: low-intensity pulsed ultrasound.

treatment of chronic coronary syndrome [5]. Moreover, some minimally invasive revascularization have been developed and clinical trials have been conducted in recent years, however, it could not prove its efficacy in the QOL of stable angina [18]. On the other hand, for the prognosis of chronic coronary syndrome, there are still major debates. In the ISCHEMIA trial with chronic coronary syndrome patients with moderate to severe ischemia, there was no difference in long-term prognosis between the intervention group treated with PCI and/or CABG and the conservative medical treatment group [3]. In the present study, the primary endpoint was

**Table 3. Clinical outcomes (MACE).**

| Adverse events (17 months) | | | | |
|---|---|---|---|---|
| | **LIPUS Group (N = 32)** | **Placebo Group (N = 29)** | **Hazard Ratio (95%CI)** | **P-value** |
| All-cause death, n (%) | 1 (3.1%) | 4 (13.8%) | 0.23 (0.027–1.922) | 0.17 |
| MACE, n (%) | 4 (12.5%) | 5 (17.2%) | 0.71 (0.192–2.587) | 0.59 |
| Cardiovascular death, n (%) | 0 (0.0%) | 1 (3.4%) | 0.00 (-) | - |
| Non-fatal myocardial infarction, n (%) | 0 (0.0%) | 1 (3.4%) | 0.00 (-) | - |
| Hospitalization due to unstable angina/heart failure, n (%) | 4 (12.5%) | 3 (10.3%) | 1.21 (0.271–5.408) | 0.80 |

MACE denotes Major Adverse Cardiovascular Events.

weekly nitroglycerin use with the aim of improving QOL in chronic coronary syndrome patients, which was dependent on subjective perception [5]. In fact, in this study, weekly nitroglycerin use, CCS score, and chest pain frequency were all significantly but equally improved in both groups, indicating that the placebo effects were involved in the subjective endpoints, as in the ORBITA trial [5]. However, importantly, this study also included objective secondary assessments such as stress myocardial perfusion imaging and prognosis. In these objective secondary endpoints, we could not reveal the statistically significant efficacy of the LIPUS therapy.

Based on these results, we consider that the efficacy of the LIPUS therapy was not shown in the present protocol. However, by analyzing myocardial perfusion imaging in individual cases, we found that some cases showed improvement in myocardial ischemia that could be regarded as a "responder". In order to assess these cases, we performed an additional post-hoc analysis after adjusting the baseline myocardial ischemic volume. We found a significant correlation between the extent of baseline myocardial ischemia and the anti-ischemic effect of the LIPUS therapy. Furthermore, the LIPUS therapy tended to improve long-term prognosis as compared with the placebo group. From these results, we consider that a next pivotal trial with a larger number of patients and objective endpoints is required for future evaluation of the LIPUS treatment.

Mechanistically, we need to consider whether the total amount of LIPUS irradiation in the present study was enough to exert sufficient therapeutic effects. In the present trial, LIPUS was applied for 3 cross-sections for 20 min each from the base to the apex of the heart 3 days in a week. The similar conditions of the LIPUS therapy have been shown to exert eNOS-mediated angiogenic effects in porcine model of chronic myocardial ischemia and mouse model of acute myocardial infarction [6, 11]. However, compared with these experimental studies with animal models, the angiogenic effects of the LIPUS therapy may not be sufficient for humans. Indeed, we also observed that repetitive LIPUS irradiation upregulated eNOS in mouse models of pressure-overload heart failure [19] and diabetes-induced heart failure with preserved ejection fraction [20]. In addition, we have recently confirmed that the LIPUS therapy on failing right ventricle (6 times in 2 consecutive weeks) was effective in a mouse model of pulmonary artery banding (PAB) and a rat model of Sugen/hypoxia (SU/Hx) [21]. We also have demonstrated the effectiveness of "whole-brain" irradiation of the LIPUS therapy in mouse models of cerebrovascular dementia [22], cerebral infarction [23], and Alzheimer's disease [22], where eNOS plays a pivotal role. Moreover, we have recently conducted a randomized, double-blind, placebo-controlled pilot trial of the LIPUS therapy for early stage of Alzheimer's disease [24]. We found that the LIPUS therapy tended to suppress cognitive impairment at 24, 48, and 72 weeks as compared with the placebo group [24]. Thus, irradiation to the entire heart, that is "whole-heart" irradiation strategy, may be considered for the next pivotal trial.

Several limitations should be mentioned for the present study. First, as discussed above, we need to define objective findings, especially myocardial flow-related data, as objective primary endpoint. Second, the number of patients was small for statistical validation. Third, as discussed above, in order to enhance the anti-ischemic effects, repetitive performance of the LIPUS therapy for the whole heart needs to be considered for the next pivotal trial. Fourth, in this study, patients with treatment-resistant angina pectoris were enrolled compared to the past studies shown above, and it is possible that the trial design was difficult to mining the effectiveness of the LIPUS therapy. Fifth, regarding the occurrence of competing risk events, only one or two deaths and discontinuation due to adverse events occurred, which was quite small. As a result, the need of using competing risk analysis has become less. All these considerations need to be included when designing the protocol of the next pivotal trial.

In summary, in the present pilot trial, we demonstrate the safety but not the additional efficacy of the LIPUS therapy for patients with refractory angina pectoris treated with optimal medical therapy. The present findings need to be further examined in the next pivotal trial with a large number of patients and modified protocols of the LIPUS therapy with objective endpoints.

## Supporting information

**S1 Checklist. CONSORT 2010 checklist of information to include when reporting a randomised trial\*.**
(DOC)

**S2 Checklist.** *PLOS ONE* **clinical studies checklist.**
(DOCX)

**S1 Fig. Representative myocardial perfusion images.** Myocardial perfusion images in a representative case in response to the LIPUS therapy.
(PDF)

**S2 Fig. Post-hoc analysis of the LIPUS therapy on objective outcome.** Correlation between the extent of baseline myocardial ischemia and that of post-treatment changes in myocardial ischemia evaluated by stress myocardial perfusion imaging. LIPUS: low-intensity pulsed ultrasound.
(PDF)

**S1 File.**
(PDF)

**S2 File.**
(DOCX)

**S3 File.**
(PDF)

**S4 File.**
(DOCX)

## Acknowledgments

We thank Katsumi Miyauchi, M.D., Ph.D and Kiyoshi Takasu, M.D., Ph.D., Juntendo University Graduate School of Medicine, Tokyo, Japan, and Hirokuni Akahori, M.D., Ph.D., Hyogo Medical University Graduate School of Medicine, Nishinomiya, Japan. We also thank Tomonori Ishii, Shoko Otomo, and Yuko Sawada, Department of Clinical Trial Implementation, Clinical Research, Innovation and Education Center, Tohoku University Hospital (CRIETO), for coordinating the trial, Koji Ikeda, Fumie Abe, Karin Ono, Satoki Kadota, Manami Hoshi, and Yoshimasa Yamazaki, CRIETO, for supporting the trial, and Airi Takagi and Takuhiro Yamaguchi, CRIETO, for statistical analysis. Appreciation is also expressed to all patients and their families who made our work possible.

## Author Contributions

**Conceptualization:** Hiroaki Shimokawa.

**Data curation:** Tomohiko Shindo, Kenta Ito, Tsuyoshi Ogata, Ryo Kurosawa, Kumiko Eguchi, Yuta Kagaya, Kenichiro Hanawa, Yuhi Hasebe, Kensuke Nishimiya, Takashi Shiroto, Jun

Takahashi, Yasuo Okumura, Teruo Noguchi, Yukio Ozaki, Hiroyuki Daida, Nobuhisa Hagiwara, Tohru Masuyama, Taishiro Chikamori, Yoshihiro Fukumoto, Kenichi Tsujita, Satoshi Yasuda.

**Formal analysis:** Tomohiko Shindo.

**Funding acquisition:** Tomohiko Shindo.

**Investigation:** Tomohiko Shindo, Kenta Ito, Kumiko Eguchi.

**Methodology:** Tomohiko Shindo.

**Project administration:** Tomohiko Shindo, Kumiko Eguchi, Hiroaki Shimokawa.

**Supervision:** Hiroshi Kanai, Satoshi Yasuda, Hiroaki Shimokawa.

**Writing – original draft:** Tomohiko Shindo.

**Writing – review & editing:** Hiroaki Shimokawa.

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
