## [Decision Letter · Decision Letter 0]

21 Mar 2023

PONE-D-22-33081A Randomized, Double-blind, Placebo-controlled Pilot Trial of Low-intensity Pulsed Ultrasound Therapy for Refractory Angina PectorisPLOS ONE

Dear Dr. Shimokawa,

Thank you for submitting your manuscript to PLOS ONE. After careful consideration, we feel that it has merit but does not fully meet PLOS ONE’s publication criteria as it currently stands. Therefore, we invite you to submit a revised version of the manuscript that thorouhgly addresses all the points raised during the review process and detailed below.

We look forward to receiving your revised manuscript.

Kind regards,

Marc W. Merx, MD

Academic Editor

PLOS ONE

Journal Requirements:

2. We note that the original protocol that you have uploaded as a Supporting Information file contains an institutional logo. As this logo is likely copyrighted, we ask that you please remove it from this file and upload an updated version upon resubmission.

Reviewers' comments:

Reviewer's Responses to Questions

**Comments to the Author**

1. Is the manuscript technically sound, and do the data support the conclusions?

Reviewer #1: Partly

Reviewer #2: Partly

2. Has the statistical analysis been performed appropriately and rigorously? 

Reviewer #1: Yes

Reviewer #2: N/A

3. Have the authors made all data underlying the findings in their manuscript fully available?

Reviewer #1: No

Reviewer #2: Yes

4. Is the manuscript presented in an intelligible fashion and written in standard English?

Reviewer #1: Yes

Reviewer #2: Yes

5. Review Comments to the Author

Reviewer #1: The manuscript addresses an interesting topic. The collected data are original and the methods employed to the analysis are rather sound. Some comments follow.

1. The data are not fully available. This is not in line with the journal's standard and, more importantly, this does not allow the reviewer to check for the correctness of the results. Publicly available data allows for the reproducibility of the work. The code used to obtain the results in the paper should also be provided.

2. The sample size is rather small. I am wondering if the assumptions underlying the employed parametric tests are met. Please, provide evidence that all the assumptions (e.g. Normality, homoschedasticity, etc.) are met.

3. The use of a competing risk models is sound overall, but should be better justified and discussed. A little-known limitation of this approach is that, for some subjects and for some time points, the sum of the subject-specific probabilities for the different event types can exceed one; this issue should be acknowledged and discussed. The very low number of events strongly limits the usefulness of the analysis: I am really doubtful on the need of this modelling as adverse events are almost null.

4. Results should be better presente. For example, the authors should show the results of Gray’s test for

equality of CIFs across groups. A plot of estimated CIFs for each cause of failure–disease combination is also required. Computing confidence intervals provides useful information about uncertainty related to parameter estimates; a

pointwise confidence interval for CIF at some fixed timepoint can be obtained using the method proposed by Choudhur (Non-parametric confidence interval estimation for competing risks analysis: application to contraceptive data. Stat Med 2002; 21: 1129–1144)

Reviewer #2: In this small pilot trial, authors investigated the efficacy and safety of a low-intensity pulsed ultrasound (LIPUS) therapy for patients with refractory angina without indication of PCI or CABG. Although this is an interesting paper, there are several issues that I am concerning about:

1. I do not understand what does it mean “patients with refractory angina without indication of PCI or CABG”. This should be clearly defined in the manuscript, as well as in the abstract.

2. In the Figure 1B follow-up was 12 months, whereas in Figures 2C and 2D, as well as in the Table 3 follow-up was 510 days? Therefore, median follow-up should be clearly defined in the manuscript, as well as in the abstract. Please, provide years or months instead of days in Figures 2C and 2D, as well as in the Table 3.

3. In the subheading “Endpoints”, the sentence “In table 2, we reported all the changes (delta) of variables between baseline and days after the initial implementation of treatment, including SDS, wall motion abnormalities at rest” is unnecessary.

4. The Table 1 is unclear. The table caption is insufficient.

Age, systolic and diastolic blood pressure, heart rate, height and ejection fraction should be presented as a whole number rather than decimals.

Percent in all tables, manuscript and abstract should also be presented as a whole number.

For continuous variables, for example: instead of “Age, years,” it should stand “Age + SD, years”; or instead of “Height, cm”it should stand “Height + SD, cm”.

All categorical variables should be written as “Variable, n (%”), for example: “Hypertension, n (%)”, etc.

Also, provide full names of all abbreviations in the table legend. This also should be applied in all tables in the manuscript (Table 2 and Table 3).

5. What does it mean “NP” in the Figure 2B? Please, provide full names of all abbreviations in all figure legends.

6. Please, provide 95% CI in figures 2C and 2D, as well as in the manuscript.

6. PLOS authors have the option to publish the peer review history of their article (what does this mean?). If published, this will include your full peer review and any attached files.

Reviewer #1: No

Reviewer #2: No

---

## [Author Response · Author response to Decision Letter 0]

4 May 2023

Responses to Reviewer 1

Manuscript: PONE-D-22-33081/R1

Authors: Shindo T, et al. 

Title: A Randomized, Double-blind, Placebo-controlled Pilot Trial of Low-intensity Pulsed Ultrasound Therapy for Refractory Angina Pectoris

The manuscript addresses an interesting topic. The collected data are original and the methods employed to the analysis are rather sound. Some comments follow.

(Response)

We would like to thank the Reviewer for the valuable comments on our work. In line with them, we have revised our manuscript. In order to facilitate the review process, we have shown the revised sentences in red in the revised manuscript.

1. The data are not fully available. This is not in line with the journal's standard and, more importantly, this does not allow the reviewer to check for the correctness of the results. Publicly available data allows for the reproducibility of the work. The code used to obtain the results in the paper should also be provided.

(Responses)

Thank you very much for this important comment. As pointed out by the Reviewer, we consent to the full disclosure of our data. In the revised manuscript, we have added this point to the Methods section as below. 

(Methods section) (Page 7, para. 1, lines 11-12)

All the clinical data obtained in this trial can be disclosed upon appropriate request.

2. The sample size is rather small. I am wondering if the assumptions underlying the employed parametric tests are met. Please, provide evidence that all the assumptions (e.g. Normality, homoschedasticity, etc.) are met.

(Responses)

Thank you very much for this important comment. Regarding the assumptions (normality, homoschedasticity), based on the following Wilcoxon test and distribution, we believe that the assumptions used here were appropriate. As shown below, t-test was used for the primary endpoint and most of secondary endpoints because it was treated as a continuous quantity, and Wilcoxon's rank sum test (Mann–Whitney U test) was performed for CCS class score.

3. The use of a competing risk models is sound overall, but should be better justified and discussed. A little-known limitation of this approach is that, for some subjects and for some time points, the sum of the subject-specific probabilities for the different event types can exceed one; this issue should be acknowledged and discussed. The very low number of events strongly limits the usefulness of the analysis: I am really doubtful on the need of this modelling as adverse events are almost null.

(Responses)

Thank you very much for this important comment. As the Reviewer pointed out, the problem of cause-specific proportional hazards is that the independence of factors must be assumed. When the probability of occurrence of the event of interest is overestimated, it happens that the sum exceeds one. In this study, we used Gray's subdistribution hazards instead of cause-specific proportional hazards to avoid such problems. However, regarding the occurrence of competing risk events, only one or two deaths and discontinuation due to adverse events occurred, which was quite small. As a result, as the Reviewer pointed out, the need of using competing risk analysis has become less. In the revised manuscript, we have discussed this point as one of the study limitations as follows;

(Discussion section) (Page 15, para. 2, lines 8-11)

Fifth, regarding the occurrence of competing risk events, only one or two deaths and discontinuation due to adverse events occurred, which was quite small. As a result, the need of using competing risk analysis has become less. 

4. Results should be better presented. For example, the authors should show the results of Gray’s test for equality of CIFs across groups. A plot of estimated CIFs for each cause of failure–disease combination is also required. Computing confidence intervals provides useful information about uncertainty related to parameter estimates; a pointwise confidence interval for CIF at some fixed timepoint can be obtained using the method proposed by Choudhur (Non-parametric confidence interval estimation for competing risks analysis: application to contraceptive data. Stat Med 2002; 21: 1129–1144).

(Responses)

Thank you very much for this valuable suggestion. In line with Reviewer's suggestion, we have performed estimation without using competing risk analysis and found that the occurrence of competing risks was slight and had little effect. Plots of all-cause mortality and MACE without competing risk analysis are shown below. We also have analyzed a pointwise confidence interval for CIF at some fixed timepoint, but the number was too small to find utility for competing risk analysis. 

• All-cause death with and without competing risk analysis

• MACE with and without competing risk analysis

• Pointwise confidence interval for CIF at some fixed timepoint

All-cause death

MACE

Finally, we again would like to thank the Reviewer for the valuable comments on our work, which have greatly helped us improve our manuscript. We sincerely hope that our revised manuscript may again be considered for publication in the Journal. 

Responses to Reviewer 2

Manuscript: PONE-D-22-33081/R1

Authors: Shindo T, et al. 

Title: A Randomized, Double-blind, Placebo-controlled Pilot Trial of Low-intensity Pulsed Ultrasound Therapy for Refractory Angina Pectoris

In this small pilot trial, authors investigated the efficacy and safety of a low-intensity pulsed ultrasound (LIPUS) therapy for patients with refractory angina without indication of PCI or CABG. Although this is an interesting paper, there are several issues that I am concerning about:

(Response)

We would like to appreciate the Reviewer’s valuable comments on our work. In line with the comments, we have revised our manuscript. In order to facilitate the review process, we have shown the revised sentences in red in the revised manuscript.

1. I do not understand what does it mean “patients with refractory angina without indication of PCI or CABG”. This should be clearly defined in the manuscript, as well as in the abstract.

(Responses)

Thank you very much for this comment. Specifically, on the premise that existing optimal medical therapy was performed, we enrolled patients in whom PCI or CABG was difficult to apply due to the lack of graftability or complexity of coronary lesions at the heart team conference. As the Reviewer pointed out, we have described this point in the revised manuscript as follows: 

(Method section) (Page 6, para. 1, lines 3-6)

Specifically, on the premise that existing optimal medical therapy was performed, we enrolled patients in whom PCI or CABG was difficult to apply due to the lack of graftability or complexity of coronary lesions at the heart team conference.

(Abstract section) (Page 3, para. 2, lines 2-5)

The patients who received optimal medical therapy without indication of PCI or CABG due to the lack of graftability or complexity of coronary lesions were enrolled.

2. In the Figure 1B follow-up was 12 months, whereas in Figures 2C and 2D, as well as in the Table 3 follow-up was 510 days? Therefore, median follow-up should be clearly defined in the manuscript, as well as in the abstract. Please, provide years or months instead of days in Figures 2C and 2D, as well as in the Table 3.

(Responses)

Thank you very much for this important comment. The median follow-up period was 452 days for the LIPUS group and 449 days for the Placebo group. In the revised manuscript, these median values have been added to the Results section as follows. Also, in line with the Reviewer’s comment, the description of time has been changed from day to month in the Figures 2C, 2D, and Table 3.

(Results section) (Page 11, lines 18-19)

The median follow-up period was 452 days for the LIPUS group and 449 days for the placebo group.

3. In the subheading “Endpoints”, the sentence “In table 2, we reported all the changes (delta) of variables between baseline and days after the initial implementation of treatment, including SDS, wall motion abnormalities at rest” is unnecessary.

(Responses)

Thank you for this suggestion. Accordingly, we have removed this sentence in the revised manuscript. 

4. The Table 1 is unclear. The table caption is insufficient. Age, systolic and diastolic blood pressure, heart rate, height and ejection fraction should be presented as a whole number rather than decimals. Percent in all tables, manuscript and abstract should also be presented as a whole number. For continuous variables, for example: instead of “Age, years,” it should stand “Age + SD, years”; or instead of “Height, cm” it should stand “Height + SD, cm”. All categorical variables should be written as “Variable, n (%”), for example: “Hypertension, n (%)”, etc. Also, provide full names of all abbreviations in the table legend. This also should be applied in all tables in the manuscript (Table 2 and Table 3).

(Responses)

Thank you very much for this valuable comment. In line with the Reviewer’s comment, in the revised manuscript, we have revised Tables 1-3.

5. What does it mean “NP” in the Figure 2B? Please, provide full names of all abbreviations in all figure legends.

(Responses)

“NP” was used to denote no statistically significant difference was found, but we have removed that in the revised manuscript as it is misleading. In addition, we have provided full names of all abbreviations in all figure legends (page 25) and table legends (pages 21-24). 

6. Please, provide 95% CI in figures 2C and 2D, as well as in the manuscript.

(Responses)

We agree with the Reviewer's comment. In the revised manuscript, we have added each 95%CI to Figures 2C and 2D in the revised figures.

Finally, thank you very much for giving us the opportunity to strengthen our manuscript with your valuable comments and queries. We believe that we have addressed the Reviewer’s comments and hope that revised manuscript is now acceptable for publication in the Journal.

---

## [Decision Letter · Decision Letter 1]

12 Jun 2023

A Randomized, Double-blind, Placebo-controlled Pilot Trial of Low-intensity Pulsed Ultrasound Therapy for Refractory Angina Pectoris

PONE-D-22-33081R1

Dear Dr. Shimokawa,

We’re pleased to inform you that your manuscript has been judged scientifically suitable for publication and will be formally accepted for publication once it meets all outstanding technical requirements.

Kind regards,

Marc W. Merx, MD

Academic Editor

PLOS ONE

Additional Editor Comments (optional):

Reviewers' comments:

Reviewer's Responses to Questions

**Comments to the Author**

1. If the authors have adequately addressed your comments raised in a previous round of review and you feel that this manuscript is now acceptable for publication, you may indicate that here to bypass the “Comments to the Author” section, enter your conflict of interest statement in the “Confidential to Editor” section, and submit your "Accept" recommendation.

Reviewer #1: All comments have been addressed

Reviewer #2: All comments have been addressed

2. Is the manuscript technically sound, and do the data support the conclusions?

Reviewer #1: (No Response)

Reviewer #2: Yes

3. Has the statistical analysis been performed appropriately and rigorously? 

Reviewer #1: (No Response)

Reviewer #2: Yes

4. Have the authors made all data underlying the findings in their manuscript fully available?

Reviewer #1: (No Response)

Reviewer #2: Yes

5. Is the manuscript presented in an intelligible fashion and written in standard English?

Reviewer #1: (No Response)

Reviewer #2: Yes

6. Review Comments to the Author

Reviewer #1: (No Response)

Reviewer #2: The manuscript is significantly improved and could be published in the present form. Additionally, please provide the full name of abbreviation LIPUS in each table.

7. PLOS authors have the option to publish the peer review history of their article (what does this mean?). If published, this will include your full peer review and any attached files.

Reviewer #1: No

Reviewer #2: No

---

## [Editor Report · Acceptance letter]

16 Jun 2023

PONE-D-22-33081R1 

A Randomized, Double-blind, Placebo-controlled Pilot Trial of Low-intensity Pulsed Ultrasound Therapy for Refractory Angina Pectoris 

Dear Dr. Shimokawa:

I'm pleased to inform you that your manuscript has been deemed suitable for publication in PLOS ONE. Congratulations! Your manuscript is now with our production department. 

Kind regards, 

on behalf of

Prof. Dr. Marc W. Merx 

Academic Editor

PLOS ONE